# Endometrial Cancer Patient-Derived Xenograft Models: A Systematic Review

**DOI:** 10.3390/jcm11092606

**Published:** 2022-05-06

**Authors:** Tomohito Tanaka, Ruri Nishie, Shoko Ueda, Shunsuke Miyamoto, Sousuke Hashida, Hiromi Konishi, Shinichi Terada, Yuhei Kogata, Hiroshi Sasaki, Satoshi Tsunetoh, Kohei Taniguchi, Kazumasa Komura, Masahide Ohmichi

**Affiliations:** 1Department of Obstetrics and Gynecology, Educational Foundation of Osaka Medical and Pharmaceutical University, 2-7 Daigakumachi, Takatsuki, Osaka 569-8686, Japan; ruri.nishie@ompu.ac.jp (R.N.); shouko.ueda@ompu.ac.jp (S.U.); shunsuke.miyamoto@ompu.ac.jp (S.M.); sosuke.hashida@ompu.ac.jp (S.H.); hiromi.konishi@ompu.ac.jp (H.K.); shinichi.terada@ompu.ac.jp (S.T.); yuhei.kogata@ompu.ac.jp (Y.K.); hiroshi.sasaki@ompu.ac.jp (H.S.); satoshi.tsunetoh@ompu.ac.jp (S.T.); m-ohmichi@ompu.ac.jp (M.O.); 2Translational Research Program, Educational Foundation of Osaka Medical and Pharmaceutical University, 2-7 Daigakumachi, Takatsuki, Osaka 569-8686, Japan; kohei.taniguchi@ompu.ac.jp (K.T.); kazumasa.komura@ompu.ac.jp (K.K.)

**Keywords:** endometrial cancer, patient-derived xenograft, PDX

## Abstract

Background: Because patient-derived xenograft (PDX) models resemble the original tumors, they can be used as platforms to find target agents for precision medicine and to study characteristics of tumor biology such as clonal evolution and microenvironment interactions. The aim of this review was to identify articles on endometrial cancer PDXs (EC-PDXs) and verify the methodology and outcomes. Methods: We used PubMed to research and identify articles on EC-PDX. The data were analyzed descriptively. Results: Post literature review, eight studies were selected for the systematic review. Eighty-five EC-PDXs were established from 173 patients with EC, with a total success rate of 49.1%. A 1–10 mm^3^ fragment was usually implanted. Fresh-fragment implantation had higher success rates than using overnight-stored or frozen fragments. Primary tumors were successfully established with subcutaneous implantation, but metastasis rarely occurred; orthotopic implantation via minced tumor cell injection was better for metastatic models. The success rate did not correspond to immunodeficiency grades, and PDXs using nude mice reduced costs. The tumor growth period ranged from 2 weeks to 13 months. Similar characteristics were observed between primary tumors and PDXs, including pathological findings, gene mutations, and gene expression. Conclusion: EC-PDXs are promising tools for translational research because they closely resemble the features of tumors in patients and retain molecular and histological features of the disease.

## 1. Introduction

Endometrial cancer is the most common cancer affecting the female reproductive system. Approximately 90% of cases have low recurrence risk, indicative of early stage and low malignancy. However, the remaining 10% of cases have poor prognoses [1,2]. Patients with advanced, high-grade, or recurrent disease require the development of breakthrough drugs.

Cell lines have been widely used in cancer research. In molecular science, they offer reliable data because of their identical gene arrangements. Although several drugs have been produced for cancer research using cell lines, drug sensitivity has not been satisfactory; some drugs are not effective in cell lines, similar to that observed in the human body [3,4,5,6]. Patient-derived xenograft (PDX) models are made from cancer tissue and are implanted directly into immunodeficient mice. PDX models have pathologic and genomic findings similar to those of the original tumors [7,8,9,10,11,12,13,14,15]. They are useful for drug discovery, identification and confirmation of biomarkers for drug sensitivity, and precision medicine (Figure 1) [6,7,16,17,18]. Trastuzumab and lapatinib were found to inhibit tumor growth in PDX models of human epidermal growth factor receptor 2 (HER2)-amplified colorectal cancer [19]. These findings were confirmed by a subsequent clinical trial [20]. PDX models of various cancers have been reported, such as those from breast [21], colon [22], stomach [23], pancreas [24], bladder [25], lung [26], kidney [27], cervix [6], endometrium [8,9,10,11,12,13,14,15] and ovary [28,29]. In this review, the data on research of endometrial cancer PDX (EC-PDX) models will be evaluated.

## 2. Materials and Methods

### 2.1. Protocol and Registration

Published articles on EC-PDX in the National Library of Medicine (PubMed) were systematically reviewed according to the Preferred Reporting Items for Systematic Reviews and Meta-analyses (PRISMA) guidelines [30]. Although reviews related to EC-PDX were searched on PROSPERO using the MeSH terms “endometrial neoplasms” and “Xenograft Model Antitumor assay”, we could not find any previous or ongoing reviews.

### 2.2. Information Sources and Search Strategies

The articles were retrieved from PubMed. The search strategy used is described as follows: (endometrial cancer [MeSH terms]) AND (antitumor assay, xenograft [MeSH terms]). Furthermore, we manually reviewed the references of all selected articles. Ryyan (http://rayyan.qcri.org, accessed on 20 March 2022) was used as the screening tool.

### 2.3. Eligibility Criteria

We screened for experiments that used EC-PDX mouse models. There were no restrictions on the number of passages of xenografts and the year of publication. The exclusion criteria were as follows: (1) xenografts using established cell lines; (2) xenografts provided with in vitro manipulation; and (3) conference proceedings, abstracts, and commentaries.

### 2.4. Study Selection

For the initial screening, the title and abstract were independently reviewed by two authors (Tomohito Tanaka and Shoko Ueda). The selected articles were read thoroughly to determine whether the entire text in the article matched the inclusion criteria. If the reviewers had conflicting views on a paper, the decision was made by a third reviewer (Masahide Ohmichi) after consultation. The reason for exclusion was specified during the second screening.

### 2.5. Data Extraction and Synthesis

Based on the selected articles, we provided a list containing the following information: (1) name of authors, (2) publication date, (3) country, (4) experimental animal used, (5) original tumor histology, (6) method for obtaining the primary tumor, (7) transplantation procedure, (8) time for transplantation, (9) fragment size, (10) site of engraftment, (11) grafting method, (12) time for tumor establishment, (13) donor patient number, (14) engraftment rate, and (15) main purpose of the study. In addition, a target was provided for drug testing. Annotations were added to the studies describing the validation methods, including preservation of histology, driver gene mutations, gene expression, copy number polymorphisms, immunohistochemistry, and proteomics.

### 2.6. Quality Assessment

A critical evaluation was performed for each selected article using the form reported by Collins et al. [31]. The evaluations were carried out based on the availability of the following data: (1) statement of ethical approval, (2) clear and detailed description of the animal model, (3) clear description of routine maintenance of the animal model, (4) preparation of the model for the experiment, (5) information about the tracked/proven tissue of origin, (6) confirmed use of donor patient xenografts, (7) histological confirmation of both the xenograft and primary tumor, and (8) information about concordance between the PDX model and the patient with respect to response to standard therapy. For each criterion, the selected studies were categorized into four sets; “yes” denoting low-risk bias, “no” denoting high-risk bias, “unclear” denoting unclear-risk bias, and “N/A” denoting not applicable.

## 3. Results

### 3.1. Study Selection

We searched for relevant studies published between 2004 and 2022. The literature search yielded 560 articles. Among these, 186 were obtained from PubMed and 374 from references. We also removed 32 duplicate articles. The inclusion and exclusion criteria were applied to the remaining 528 articles, and 74 were selected for full-text reading. Post screening, eight articles were selected and included in this systematic review. The flowchart in Figure 2 shows the literature search and study selection process. 

### 3.2. General Features of EC-PDX Models

Table 1 and Table 2 present the characteristics of the studies included in this review. Studies from seven countries were included: Spain, USA, Belgium, Norway, China, Australia, and South Korea. Three different animal models, including nude, non-obese diabetic (NOD), severe combined immunodeficient (SCID), and NOD SCID gamma (NSG) mice, were used. Several histological types of primary tumors were reported, including endometrioid carcinoma, serous carcinoma, clear cell carcinoma, carcinosarcoma, and undifferentiated carcinoma. In most studies, the tumors were obtained from surgically resected specimens. The time between surgery and animal implantation was described in five studies, and ranged from 0 (immediately) to 5 h. The tissues were stored overnight at 4 °C, and were frozen in one study. The implanted tissue fragment size ranged from 1–10 mm^3^, and two articles implanted a cell suspension, post centrifugation. The most common transplantation site was subcutaneous tissue, followed by the orthotopic endometrial cavity and the subrenal capsule. In most cases, the graft was directly implanted through a skin incision and/or transabdominally. In two studies, minced tumor fragments were injected transvaginally into the endometrial cavity. Five studies mentioned that the latency period until tumor growth varied from 2 weeks to 13 months. The number of donor patients ranged from 1 to 64. Seven studies reported their success rates, which ranged from 36.4–100%. A total of 85 EC-PDXs were established from 173 patients with EC, with an overall success rate of 49.1%.

The validation methods and parameters used to demonstrate the characteristics of the PDXs and donor patient tumors are presented in Table 3. Histological comparisons between PDXs and original tumors were reported in seven studies. Driver gene mutations and gene expression were described in three articles. Copy number variations were reported in two articles. However, proteomic analyses were not conducted in most studies. Immunohistochemical analysis was performed in three studies, using p53, ER, PR, and Ki67 antibodies in most cases.

### 3.3. Quality Assessment

The online model validation tool [31] was used to further assess the PDX models (Figure 3). Most studies provided an ethics statement, model details, routine maintenance of the model, and confirmation of PDXs. Several studies lacked reports on further preparation of the model, tracking/preparation of the tissue model, or histological comparisons among primary tumors. The concordance to treatment response was described poorly in most studies.

## 4. Discussion

The success rate of EC-PDX development was 49.1%. This rate did not decrease during subcutaneous implantation in nude mice. Similar characteristics were observed between primary tumors and PDXs, including pathological findings, gene mutations, and gene expression. 

### 4.1. Success Rate and Transplantation Method

The success rate depended on several factors, including the stage and histology of the primary tumor, fragment size, animal model choice, and the transplantation site [6]. Usually, the success rate is higher when immunodeficient mice are used; however, there are multiple mice suitable for several cancers [6]. In gastric cancer, patients with advanced disease have higher IgG levels than those with early disease, and IgG is expected to play an important role in tumor proliferation and infiltration. SCID mice lose existing B and T cells. In contrast, although nude mice lose their B cell function, the number of B cells remains normal. Thus, nude mice are more suitable for use as PDX models for gastric cancer [32,33]. Shin et al. insisted that PDX models in gynecological cancer did not correspond to immunodeficienct grades, and PDX using nude mice reduced costs [15]. The success rate of the EC-PDX model using nude mice was not low in their study.

EC-PDX models are established with common histological types, including endometrioid carcinoma, clear cell carcinoma, serous carcinoma, and carcinosarcoma; however, the success rate depends on the tumor grade. Bonazzi et al. reported that successful engraftments were only obtained for histological grades 2 and 3 tumors, but not for grade 1 tumors. They also reported a higher implantation success rate for fresh fragments than for frozen or overnight-stored fragments [14].

Subcutaneous implantation is most common for EC-PDXs because it is easy to perform and confirm tumor growth [8,10,13,14,15]. However, metastases rarely occur in subcutaneously implanted tumors. Usually, the engraftment rate is higher in models with transplantation into the subrenal capsule than in those into other transplantation sites; however, it is difficult to perform procedures and confirm tumor growth in models with subrenal capsule transplants. Orthotopic models reproduce tumor conditions accurately. There are two reports of orthotopic models of endometrial cancer, representing tumor growth in the uterine horn. Metastasis was observed in most of those models [8,11]. 

### 4.2. Comparison of Original Tumors and PDXs

In most studies, the pathological characteristics, including structural and cytological features, between primary tumors and PDXs were similar [8,9,10,11,12,13,14]. These findings were preserved after several passages [10,14]. Several studies have performed immunohistochemical analyses of p53, Ki67, estrogen receptor, and progesterone receptor. Similar staining patterns were observed between primary tumors and PDXs [8,9,10]. Recently, DNA and RNA sequencing have been used in EC-PDXs. In a study by Zhu et al., DNA and RNA sequencing were performed to compare the original tumors with F4 PDXs in two high-grade endometrial cancers. Most mutations in the primary tumors and the PDXs were similar. The mutation frequencies showed a significant linear correlation. The RNA sequences also showed a significant linear correlation with gene expression [13]. In a study by Depreeuw et al., whole exon sequences were analyzed in grade 1 and 3 endometrioid carcinomas without microsatellite instability (MSI)-related gene and DNA polymerase epsilon (POLE) mutations. Most mutations between the primary tumors and the PDXs were similar. On average, 90% of the genome had the same copy count between the primary tumor and the PDX [10]. In the study by Bonazzi et al., whole exome sequencing was performed on endometrial cancers with four common molecular subtypes, including POLE mutations, mismatch repair deficiency (MMRd), p53 mutations, and no specific molecular profile. Interestingly, they focused on the MMRd mutation subtype, because it is expected to accumulate changes during passages based on the loss of DNA mismatch repair. They found that mutational heterogeneity was minimal in non-MMRd models, but was more frequent in MMRd models. In the p53 mutation subtype, the total number of somatic mutations was consistent between the primary tumors and PDXs [14]. Cybula et al. focused on breast cancer gene (BRCA)-mutated ovarian serous carcinomas. They expected changes to accumulate during passages based on the DNA repair deficiency caused by BRCA mutations, and they performed genomic analysis focused on single nucleotide polymorphisms (SNPs). In their analysis, the PDXs remained largely stable throughout propagation; however, some marginal genetic drift occurred at the time of PDX initiation. They also found several genetically unstable PDXs that may be associated with DNA repair deficiency due to BRCA mutations [29]. 

### 4.3. Implication for Further Research and Research Practice

Complete surgical resection is the most effective therapy for endometrial cancer. However, this treatment is not an option for some patients with advanced or recurrent disease [1,2]. Thus, other therapies and precision medicine are needed for these patients. Established cell lines have been used for cancer research of many types of cancer, but these cells cannot simulate the heterogeneity of primary tumors [34]. PDX models may overcome this problem; similar characteristics were observed between primary tumors and PDXs, including pathological findings, gene mutations, and gene expression [8,9,10,11,12,13,14,15].

#### 4.3.1. Drug Repositioning

Repositioning of existing drugs previously approved by the FDA reduces the costs and barriers associated with clinical trials [35]. The primary purpose of these drugs was not cancer therapy. In ovarian cancer, several repositioned drugs have been evaluated using PDX models [36,37,38,39,40,41,42,43,44,45,46].

#### 4.3.2. Precision Medicine

Compared with cell lines, fresh tumor tissue shares the same genetic profile as the human body. PDX models also maintain most genetic features of the primary human tumors. In hepato-pancreato-biliary cancer, the response to different drugs was similar between patients in clinical trials and PDX models [47]; PDX models could be useful for preclinical evaluation to select suitable drugs for treatment.

#### 4.3.3. Mini-PDX Models

PDX models are suitable for precision medicine because they possess similar characteristics and drug sensitivity to the primary tumors [47]. However, these models require several months for tumor growth. “Mini-PDX” is an in vivo drug sensitivity test developed to overcome this problem, requiring only 7 days to estimate drug sensitivity. Briefly, microencapsulated tumor cells are subcutaneously implanted into mice, and the mice are treated with an anticancer drug. Drug sensitivity is estimated by measuring tumor cell proliferation in the capsule [48].

#### 4.3.4. PDX Models and Co-Clinical Trials

An “avatar” is a PDX model that receives the same anticancer agent as the donor patient received. In co-clinical trials, antitumor drugs are administered to patients with certain gene mutations and PDX models with similar gene mutations. The purpose of an “avatar” is to optimize treatment strategies in clinical trials to identify the best treatment strategy for patients [49].

#### 4.3.5. Identifying Tumor Biomarkers

PDX models help to determine useful molecular biomarkers related to drug sensitivity or drug resistance and patient prognosis. In colorectal cancer, dual HER2 blockade with trastuzumab and lapatinib led to inhibition of tumor growth in PDXs of HER2-amplified tumors [19]. A phase 2 clinical trial demonstrated the effectiveness of this therapy in treatment-refractory patients with HER2-positive metastatic colorectal cancer [20].

#### 4.3.6. Humanized PDX Models for Immunotherapy

One of the most important PDX models may be the humanized mouse for the development of immunotherapies [17]. The NSG and NOG mouse strains are suitable for creating humanized mice because they lack natural killer (NK) cells. Peripheral blood lymphocytes (PBLs) [50], CD34+ human hematopoietic cells [51], or bone marrow-liver-thymus (BLT) tissue [52] is usually used as the source of human immunological cells. After irradiation of immunodeficient mice, PBLs or CD34+ cells are transplanted intravenously, intraperitoneally, or via another route. Alternatively, a piece of BLT can be implanted into the subrenal capsule of immunodeficient mice that received prior irradiation. Thus, the tumor fragments are implanted into humanized mice with a human immune system. The antitumor immune response can then be investigated in humanized PDX models.

## 5. Limitations

The studies reviewed have certain limitations that should not be overlooked. First, the sample size was relatively small. Second, the methods and calculation of results were not standardized. For example, the number of mice used and calculation of the success rate varied. Therefore, further investigation is required to confirm our results.

## 6. Conclusions

Subcutaneous implantation of 1–10 mm^3^ fragments into nude mice may be suitable for EC-PDXs; however, orthotopic implantation with minced tumor cell injection is better for metastatic models. EC-PDX is a promising tool for translational research because it closely resembles the tumor features of patients and retains the molecular and histological features of the disease.

## Figures and Tables

**Figure 1 jcm-11-02606-f001:**
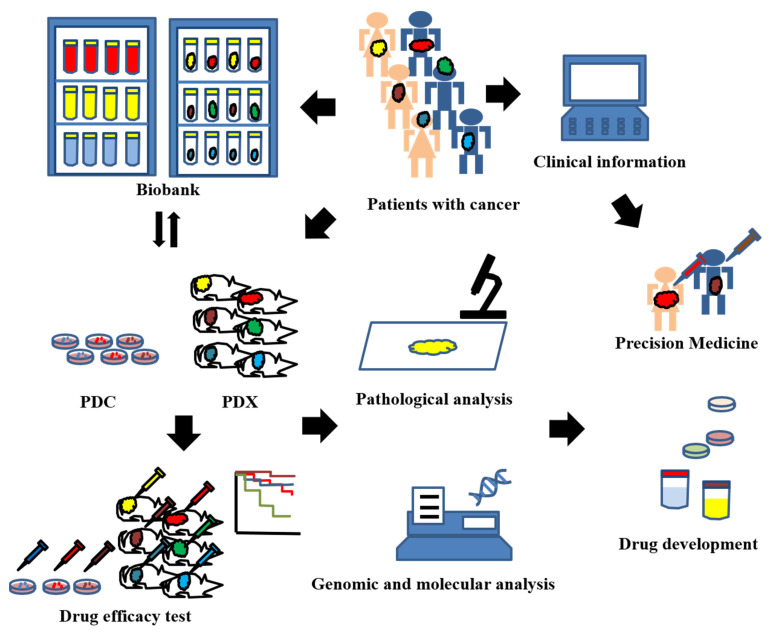
Schematic for the use of patient-derived xenograft (PDX) models. PDX models can be created by grafting the tissue obtained by surgery or biopsy into immunodeficient mice. Patient-derived cells (PDCs) are also created from tumors. All materials and information from cancer patients and PDX models are stored in biobanks and data banks. The materials include all samples obtained from patients or PDX models, such as blood, urine, discharge, and tumors. The information also includes clinicopathological, genomic analysis, and drug sensitivity data. These materials and information in biobanks and databanks are intended for use in precision medicine and the development of anticancer agents; this platform allows many researchers to share all types of information and conduct experiments with PDXs that reflect the characteristics of the primary tumor.

**Figure 2 jcm-11-02606-f002:**
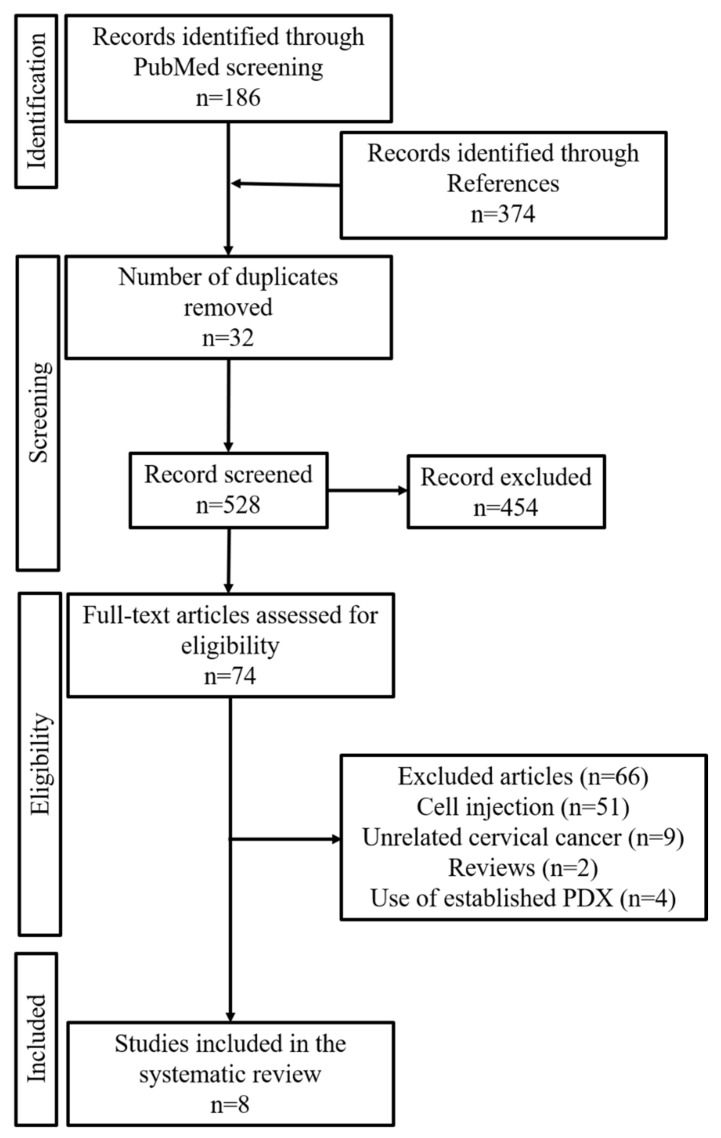
Flowchart showing the results of the search process.

**Figure 3 jcm-11-02606-f003:**
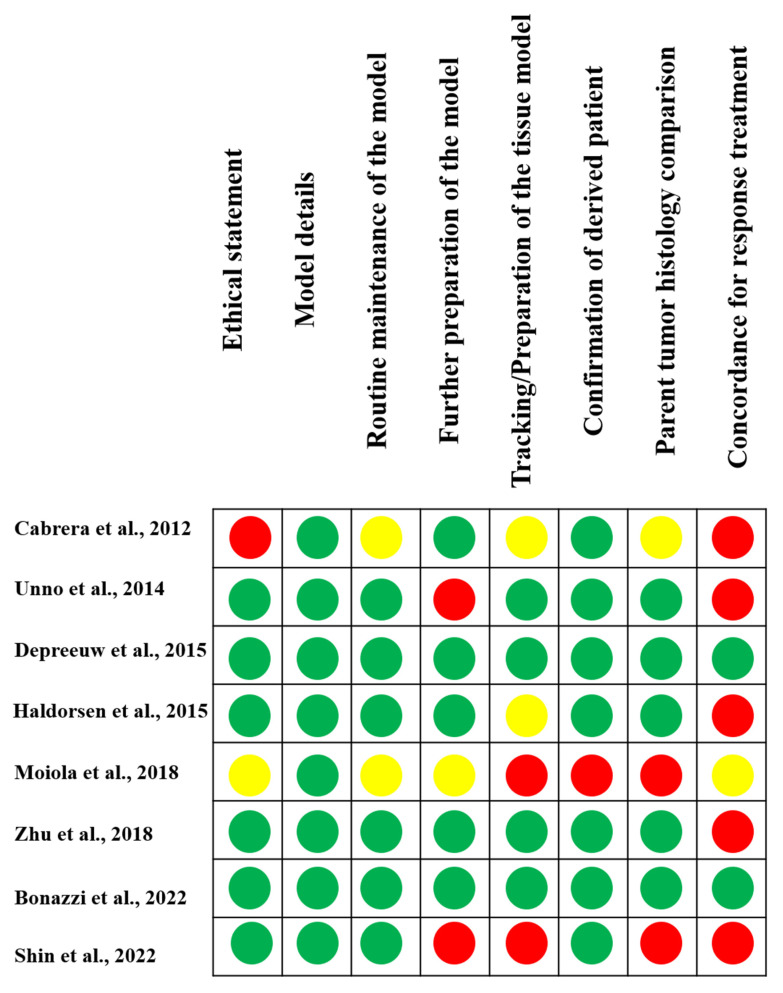
Quality assessment of the studies included in this systematic review. Green circles indicate studies that reported the evaluated item (low risk of bias); red circles indicate studies that did not report the evaluated item (high risk of bias); and yellow circles indicate studies that did not define or only partially reported the evaluated item [8,9,10,11,12,13,14,15].

**Table 1 jcm-11-02606-t001:** Characteristics of endometrial cancer patient-derived xenograft models.

Author, Year	Country	Animal Model	Histology	Type of Procedure for Obtaining the Tumor	Aim of the Study
Cabrera et al., 2012 [8]	Spain	Nude	EEC	Surgery	Evaluate the PDX method
Unno et al., 2014 [9]	USA	NSG	EEC, SEC, CCEC, and UCS	Surgery	Evaluate the PDX method
Depreeuw et al., 2015 [10]	Belgium	Nude	EEC, SEC, CCEC, and UDC	Surgery	Evaluate the PDX model
Haldorsen et al., 2015 [11]	Norway	NSG	EEC	Biopsy	Imaging evaluation using PDX model
Moiola et al., 2018 [12]	Spain	Nude or NSG	EEC, SEC, CCEC, UCS, and others	Surgery	PDX cohort
Zhu et al., 2018 [13]	China	NOD/SCID	EEC, SEC, CCEC, and UCS	Surgery	Evaluate the PDX model and drug evaluation
Bonazzi et al., 2022 [14]	Australia	NSG	EEC, SEC, CCEC, and UCS	Surgery	Evaluate the PDX model and drug evaluation
Shin et al., 2022 [15]	Korea	Nude	EEC, SEC, CCEC, and UCS	Surgery	Evaluate the PDX model

SCID, severe combined immunodeficiency; NOD, non-obese diabetic; NSG, NOD/SCID/IL2rg null; EEC, endometrioid endometrial carcinoma; SEC, serous endometrial carcinoma; CCEC, clear cell endometrial carcinoma; UCS, uterine carcinosarcoma; PDX, patient-derived xenograft.

**Table 2 jcm-11-02606-t002:** Characteristics of endometrial cancer patient-derived xenograft models.

Author, Year	Time between Surgery and Implantation	Fragment Size	Site of Transplantation	Method of Graft	Mean Latency	Number of Donor Patients	Engraftment Rate (%)
Cabrera et al., 2012 [8]	Immediately	1 mm^3^	Subcutaneous	Direct	N.I.	2	100 (2/2)
	Immediately	Crumbled	Uterine cavity	Injection	62.7 d	2	100 (2/2)
Unno et al., 2014 [9]	N.I.	1.5 mm × 1.5 mm	Renal capsule	Direct	N.I.	11	36.4 (4/11)
Depreeuw et al., 2015 [10]	Within 4 h	8–10 mm^3^	Subcutaneous	Direct	1.5–9 mo	40	60 (24/40)
Haldorsen et al., 2015 [11]	N.I.	Cell suspension	Uterine cavity	Injection	3–4 mo	1	100 (1/1)
Moiola et al., 2018 [12]	N.I.	Small tissue fragment	Orthotopic	Direct	1–5 mo	64	N.I.
	N.I.	5–10 mm^3^	Subcutaneous	Direct	2–3 mo	40	N.I.
	N.I.	8–10 mm^3^	Subcutaneous	Direct	3–5 mo	15	N.I.
	N.I.	Cell suspension	Orthotopic	Direct	3–13 mo	5	N.I.
Zhu et al., 2018 [13]	Within 5 h	1 × 1.5 × 1.5 mm^3^	Subcutaneous	Direct	2–11 wk	18	50 (9/18)
	Within 5 h	1 × 1.5 × 1.5 mm^3^	Renal capsule	Direct	4–10 wk	16	62.5 (10/16)
Bonazzi et al., 2022 [14]	Within 4 h	1–2 mm^3^	Subcutaneous	Direct	N.I.	32	61 (13/32)
	4 °C overnight	1–2 mm^3^	Subcutaneous	Direct	N.I.	11	27 (3/11)
	Viably Frozen	1–2 mm^3^	Subcutaneous	Direct	N.I.	11	18 (2/11)
Shin et al., 2022 [15]	Immediately	3 mm^3^	Subcutaneous	Direct	6 mo	31	56 (17/31)

N.I., no information; d, day; wk, weeks; mo, months.

**Table 3 jcm-11-02606-t003:** Validation methods and parameters used to demonstrate that PDXs resemble their donor patient tumors in the eight studies that explored PDX models.

Author, Year of Publication	Histology	Driver Gene Mutation	Gene Expression	Copy Number Variation	Proteomics	Immunohistochemistry	Other
Cabrera et al., 2012 [8]	Yes	No	No	No	No	p53, ER, PR, Ki67, E-cadherin, MSH2, MLH1, MSH6	No
Unno et al., 2014 [9]	Yes	No	No	No	No	p53, ER, PR, Ki67, CD31, cytokeratin, vimentin, E-cadherin, PTEN, uPA, uPAR	No
Depreeuw et al., 2015 [10]	Yes	Yes	Yes	Yes	No	ER, PR, vimentin, MLH, MSH2, cytokeratin	PI3K/mTOR and MEK inhibitor
Haldorsen et al., 2015 [11]	Yes	No	No	No	No	No	No
Moiola et al., 2018 [12]	Yes	No	No	No	No	No	No
Zhu et al., 2018 [13]	Yes	Yes	Yes	No	No	No	No
Bonazzi et al., 2022 [14]	Yes	Yes	Yes	Yes	No	No	POLE, MMRd, p53 and HRD
Shin et al., 2022 [15]	No	No	No	No	No	No	No

ER, estrogen receptor; PR, progesterone receptor; MSH, MutS homolog; MLH, MutL homolog; PTEN, phosphatase and tensin homolog; uPA, urokinase-type plasminogen activator; uPAR, urokinase-type plasminogen activator receptor; PI3K, phosphoinositide 3-kinase; mTOR, mammalian target of rapamycin; MEK, mitogen-activated protein kinase; POLE, DNA polymerase epsilon; MMRd, mismatch repair deficiency; HRD, homologous recombination deficiency.

## Data Availability

The data presented in this study are available upon request from the corresponding author.

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
