# Peer review of "Endometrial Cancer Patient-Derived Xenograft Models: A Systematic Review"

_jcm, 2022, doi:10.3390/jcm11092606_

Round 1

Reviewer 1 Report

This manuscript is well documented and organized.

Recently, PDX model is important to the cancer research.
The aim of this manuscript was to review the articles on PDX of endometrial cancer (EC-PDX) and verify the methodology and outcomes.
This manuscript reviewed several EC-PDX-related paper and provided clear information about EC-PDX.
This well documented and organized manuscript would be useful for future EC-PDX study. 
To favor for readers, a graphic model of PDX model in preclinical and clinical application should be provided.

Author Response

Point-by-Point Responses to Reviewers

We appreciate the time and effort of the editor and referees in reviewing our manuscript. We have addressed all of the issues indicated in the review report and hope that the revised version meets the journal's requirements for publication.

Response to Comments from Reviewer 1:

Comment 1:

Recently, PDX model is important to the cancer research.
The aim of this manuscript was to review the articles on PDX of endometrial cancer (EC-PDX) and verify the methodology and outcomes.
This manuscript reviewed several EC-PDX-related paper and provided clear information about EC-PDX.
This well documented and organized manuscript would be useful for future EC-PDX study. 
To favor for readers, a graphic model of PDX model in preclinical and clinical application should be provided.

Response:

According to your suggestion, we added a graphic model of PDX in preclinical and clinical application, which is now Figure 1 in the revised manuscript. (page 2, Figure 1)

Reviewer 2 Report

The article “Endometrial cancer patient-derived xenograft models: A systematic review” is a great addition to the available scientific literature on EC-PDX as a promising tool for translational research, it closely resembles the features of tumors in patients and retains the molecular and histological features of the disease.

The study shows that the success rate of EC-PDX development is 49.1 %. This rate did not decrease during subcutaneous implantation in nude mice. Similar characteristics were observed between primary tumors and PDX, including pathological findings, gene mutations, and expression. Moreover, subcutaneous implantation of 1–10 mm3 fragments into nude mice may be suitable for EC-PDX; however, orthotopic implantation with minced tumor cell injection is better for metastatic models.

These findings highlights that EC-PDX is a promising tool for translational research because it closely resembles the tumor features of patients and retains the molecular and histological  features of the disease.

Author Response

We appreciate your time and effort in reviewing our manuscript.

Reviewer 3 Report

Thank you for inviting me to review this manuscript. I believe the manuscript can contribute to our current understanding of PDX models, there is some further justification and content needed:

  • Restructure the abstract so it becomes more informative on the outcomes. Also reconsider the rationale of using PDX. The rationale is hardly that more researchers are doing it, but please communicate the rationale of higher translational value of PDX in the abstract.

  • Figure 1: Remove the n=0 categories from the flowchart

  • In table 1 it is unclear, what the abbreviation NSG stands for - please add in the legend. Furthermore the histology abbreviations are highly unusual: usually endometrioid EC is reffered to as EEC, serous as SEC, ...

  • Table 1: Type of procedure for obtaining material - subcutaneous PDX: this does not seem to be correct, please revise to provide only, what the type of procedure was

  • Table 2: Please report the engraftment rate in a unified manner: some include the number of cases where engraftment is present and others do not - can you please unify this?

  • Prior to Shin et al in Table 2, there are two lines stating:

11

27 (3/11)

11

18 (2/11)

 without further details. What do these lines mean? Is it a different study, is it common to Bonazzi et al?

  • Please add information on implications on further research and research practice as part of the discussion.

Author Response

Point-by-Point Responses to Reviewers

We appreciate the time and effort of the editor and referees in reviewing our manuscript. We have addressed all of the issues indicated in the review report and hope that the revised version meets the journal's requirements for publication.

Response to Comments from Reviewer 3:

Comment 1:

Thank you for inviting me to review this manuscript. I believe the manuscript can contribute to our current understanding of PDX models, there is some further justification and content needed: Restructure the abstract so it becomes more informative on the outcomes. Also reconsider the rationale of using PDX. The rationale is hardly that more researchers are doing it, but please communicate the rationale of higher translational value of PDX in the abstract.

Response:

According to your suggestion, we revised the abstract to include rationale for using PDX and more information about the reviewed studies. (page 1, line 18-43)

Comment 2:

Figure 1: Remove the n=0 categories from the flowchart

Response:

According to your suggestion, we removed the “n=0 category” from the flowchart, which is now Figure 2 in the revised manuscript. (page 4, Figure 2)

Comment 3:

In table 1 it is unclear, what the abbreviation NSG stands for - please add in the legend. Furthermore the histology abbreviations are highly unusual: usually endometrioid EC is reffered to as EEC, serous as SEC, ...

Response: These abbreviations have been updated according to your recommendations and defined in revised Table 1. (page 5, table 1)

Comment 4:

Table 1: Type of procedure for obtaining material - subcutaneous PDX: this does not seem to be correct, please revise to provide only, what the type of procedure was

Response: The entries in this column of Table 1 have been changed to either “Surgery” or “Biopsy” according to the study in question. (page 5, table 1)

Comment 5:

Table 2: Please report the engraftment rate in a unified manner: some include the number of cases where engraftment is present and others do not - can you please unify this?

Response: This has been corrected according to your suggestion. (page 5, table 2)

Comment 6:

Prior to Shin et al in Table 2, there are two lines stating:

 without further details. What do these lines mean? Is it a different study, is it common to Bonazzi et al?

Response: We apologize for this oversight on our part. These were part of the Bonazzi et al. 2022 study, and both entries were corrected in the table. (page 5, table 2)

Comment 7

Please add information on implications on further research and research practice as part of the discussion.

Response: This information has been added as section 4.3 in the discussion and includes subsections on drug repositioning, precision medicine, mini-PDX models, PDX models and co-clinical trials, identifying tumor biomarkers, and humanized PDX models for immunotherapy. (page 8, line 240-290)

Round 2

Reviewer 3 Report

My concerns have been anwsered. Thank you for this extensive revision, I believe the manuscript to be a valuable contribution to the current body of knowledge on this topic.